# Assessing Physical Activity and Sedentary Behavior under Free-Living Conditions: Comparison of Active Style Pro HJA-350IT and ActiGraph^TM^ GT3X+

**DOI:** 10.3390/ijerph16173065

**Published:** 2019-08-23

**Authors:** Shohei Yano, Mohammad Javad Koohsari, Ai Shibata, Kaori Ishii, Suzanne Mavoa, Koichiro Oka

**Affiliations:** 1Institute for Sport Sciences, Waseda University, Saitama 359-1192, Japan; 2Faculty of Sport Sciences, Waseda University, Saitama 359-1192, Japan; 3Behavioural Epidemiology Laboratory, Baker Heart and Diabetes Institute, Melbourne, VIC 3004, Australia; 4Melbourne School of Population and Global Health, The University of Melbourne, Melbourne, VIC 3010, Australia; 5Faculty of Health and Sports Sciences, University of Tsukuba, Ibaraki 305-8574, Japan

**Keywords:** accelerometers, accelerometry, objective assessment, sitting time, activity monitors

## Abstract

Various accelerometers have been used in research measuring physical activity (PA) and sedentary behavior (SB). This study compared two triaxial accelerometers—Active style Pro (ASP) and ActiGraph (AG)—in measuring PA and SB during work and nonwork days in free-living conditions. A total of 50 working participants simultaneously wore these two accelerometers on one work day and one nonwork day. The difference and agreement between the ASP and AG were analyzed using paired *t*-tests, Bland–Altman plots, and intraclass coefficients, respectively. Correction factors were provided by linear regression analysis. The agreement in intraclass coefficients was high among all PA intensities between ASP and AG. SB in the AG vertical axis was approximately 103 min greater than ASP. Regarding moderate-to-vigorous-intensity PA (MVPA), ASP had the greatest amount, followed by AG. There were significant differences in all variables among these devices across all day classifications, except for SB between ASP and AG vector magnitude. The correction factors decreased the differences of SB and MVPA. PA time differed significantly between ASP and AG. However, SB and MVPA differences between these two devices can be decreased using correction factors, which are useful methods for public health researchers.

## 1. Introduction

Physical inactivity and excessive sitting have been shown to be associated with increased risks of mortality and chronic diseases [1,2]. In particular, sedentary behavior (SB) has been found to be associated with several health outcomes, such as type 2 diabetes, cardiovascular diseases, and cancer, independent of moderate-to-vigorous-intensity physical activity (MVPA) [3,4]. Regular physical activity (PA) also provides numerous health benefits, including a reduced risk of chronic diseases [5]. Walking or jogging, which can be assessed by step counts [6], is one of the most common means of undertaking PA.

Accurate measurement of PA and SB is an important step in developing strategies to promote an active lifestyle. The two common methods of measuring PA/SB include subjective assessment through self-reported questionnaire and objective assessment using accelerometers. The self-reported approach has been commonly used in previous studies [7,8]. However, this approach has some limitations, such as recall bias [9] and social desirability bias [10]. More recently, accelerometry-based activity monitors have been widely applied to objectively assess PA levels [11,12]. An accelerometer can assess PA volume, intensity, and frequency [13] with fewer measurement errors and greater accuracy than subjective assessments [14].

The number of research studies using accelerometer devices to assess PA and SB has increased over the last few years. Several types of accelerometer devices such as ActiGraph (Actigraph Corporation, Pensacola, FL, USA), Actical (Phillips Respironics, Bend, OR, USA), Actiheart (CamNtech Ltd., Cambridge, UK), activPAL (Pal Technologies Ltd., Glasgow, UK), GENEActive (Unilever Discover, Colworth, UK), and Lifecorder (Suzuken, Co. Ltd., Nagoya, Japan) have been used. Each device has different features, such as algorithms, wearing positions (e.g., hip, wrist, and thigh), and settings (e.g., epoch, sampling rate, and nonwearing time definition). Such variations may result in different accelerometers assessing different amounts of PA and SB [15,16,17]. Moreover, even different generations of a device from the same brand can lead to different PA and SB results [18,19,20]. A study comparing two different generations of the same device showed a significant difference between ActiGraph GT1M and GT3X+ in SB (about 25 min/day), light-intensity PA (LPA) (about 31 min/day), and moderate PA (about 2 min/day) under free-living conditions [20]. In comparison studies between other devices, a study in the United States showed that the ActiGraph calculated significantly more time in light (+16.3%), moderate (+2.8%), and vigorous (+0.4%) activity than the Actical device under free-living conditions [15]. In addition, choosing different cut-off points between SB, LPA, and MVPA in accelerometers may cause differences in their outputs. For instance, a study found the differences between ActiGraph and Lifecorder devices to be dependent on the choice of cut-off points in ActiGraph [16]. Several previous studies have compared the PA and SB measures assessed by different types of accelerometer devices [21,22,23,24,25,26]. These comparison studies are important to assist in interpretation of findings from previous studies that used different types of accelerometers. They also can shed a light on choosing a suitable device for future studies. 

One of the most-commonly used activity monitors for research purposes is the ActiGraph device [13]. ActiGraph devices have been shown to provide accurate estimates of PA in free-living environments [27]. These devices have been used in several large-scale epidemiological studies across the world [28,29]. A relatively new accelerometer device called the Active style Pro (HJA-350IT) was developed in 2008 (Omron Healthcare Company; Kyoto, Japan). The triaxial Active style Pro can distinguish between ambulatory and nonambulatory activities and has high validity for estimating SB, LPA, and MVPA compared with the Douglas bag method of estimating energy expenditure under laboratory conditions [30,31]. Recently, a study showed that the Active style Pro had high validity for estimating energy expenditure compared to doubly labeled water methods under free-living conditions [32]. Another study showed that the Active style pro had high validity for estimating walking activities in LPA to MVPA compared with the Douglas bag method of measuring activity intensity (metabolic equivalents/min: METs) under laboratory conditions [33]. Thus, a growing number of studies have used Active style Pro accelerometers [34,35,36]. However, few studies have compared the Active style Pro to other accelerometers regarding outcomes in measuring various time intensities under free-living conditions. A recent study comparing behavior outputs between the Active style Pro and ActiGraph reported a significant difference in SB between these two devices [37]. Although this study chose the normal and low-frequency settings using the vertical axis when comparing ActiGraph outcomes to Active style Pro outcomes [37], the difference of SB between the Active style Pro and ActiGraph using the vector magnitude setting has not yet been examined. In addition, to our knowledge, no study has compared MVPA and LPA with the Active style Pro and the ActiGraph.

Therefore, this study aimed to compare the Active style Pro with the vertical axis and vector magnitude ActiGraph (GT3X+) in measuring various PA and SB intensities during work and nonwork days in free-living conditions. Additionally, for differences between the Active style Pro and ActiGraph in each intensity activity, we aimed to calculate correction factors and use these to compare the results.

## 2. Materials and Methods

### 2.1. Study Participants and Procedures

Data were collected from 50 participants recruited from several workplaces in Tokyo, Japan. Eligible participants were recruited based on the following inclusion criteria: healthy, aged 20 years or older, and full-time workers (who work at least five days a week and 8 h per work day). The participants included 35 staff members working at a hospital, 5 manual laborers at a factory, 1 system engineer at a company, and 9 researchers and staff at a university. Participants wore two accelerometers (an Active style Pro and an ActiGraph accelerometer) simultaneously on their right and left hip, with the placement of each device randomly determined to remove placement bias. They were instructed to wear these while awake on a total of two days (one work day and one nonwork day), without sleeping and avoiding water-based activities such as bathing and swimming, because there are different activity patterns between work and nonwork days [38,39]. Ten or more hours of accelerometer wear time per day was considered valid. An instructional document for wearing the monitors with a diary column to record the wear time was handed to each participant. The procedures were reviewed and approved by the Academic Research Ethical Review Committee at Waseda University, Japan. Each participant provided written informed consent prior to data collection.

### 2.2. Instrument and Data Management

*Devices*. The Active style Pro (HJA-350IT) is a triaxial accelerometer. It was set to calculate the average intensity of movements every 60 s for analysis. The locomotion versus nonlocomotion algorism of the Active style Pro device was used to estimate the intensity of PA [30]. The detailed algorithm in Active style Pro’s accelerometer has high validity and has been tested under free-living conditions using the doubly labeled water method and under laboratory conditions using the Douglas bag method [30,31,32,37]. The ActiGraph (GT3X+) is a small lightweight triaxial activity monitor that provides data on PA, including activity counts, energy expenditure (kcal), steps, and activity intensity (METs). The device can be hip- or wrist-worn (in this study, hip-worn) on an elastic belt and activity is measured across three perpendicular planes. In this study, a specified sampling rate was initialized to set raw data at 30 Hz. The data from 60 s epochs with the normal setting were used in the analysis [40]. The low-frequency extension setting, which increases the sensitivity of measuring small movements, was not used. Nonwear time was recorded in the diary column by participants. Participants wore one ActiGraph device and data from the vertical axis and the vector magnitude settings were downloaded. We sought to determine whether the different axial settings in ActiGraph provide similar outcomes of measuring PA and SB in the Active style Pro to compare the outcomes between different studies using ActiGraph and Active style Pro. Thus, two devices (Active style Pro and ActiGraph) and three datasets (Active style Pro, ActiGraph with the vertical axis, and ActiGraph with the vector magnitude) were used in this study. 

*Physical activity and sedentary behavior*. The accelerometers measured time in various intensities of PA and steps on work and nonwork days. The daily time spent on SB (≤1.5 METs) [41], LPA (>1.5 to <3.0 METs), and MVPA (≥3.0 METs) were calculated. PA outcomes were derived using each original method in Active style Pro and ActiGraph using the vertical axis and vector magnitude. To classify accelerometer output data into different SB and PA intensity categories, the following cut-offs were used: SB was defined as <100 counts/min for the vertical axis [42] and <200 counts/min [43] for the vector magnitude in ActiGraph, and 1.5 METs in Active style Pro; PA levels, including LPA and MVPA, were defined as ≥100 to 1951 counts/min and ≥1952 counts/min for the vertical axis, ≥200 to 2689 counts/min and ≥2690 counts/min for the vector magnitude in ActiGraph using Freedson adult activity cut points for the vertical axis [44], and Sasaki activity cut points for vector magnitude [45] in ActiGraph and MET-based cut points in Active style Pro (LPA: 1.6–2.9 MVPA; ≥3.0 METs). Freedson and Sasaki cut points of MVPA are widely used in research using hip-worn ActiGraph accelerometers [44,45]. Sasaki activity cut points are commonly used for vector magnitude data [46,47]. The Active style Pro used its original MET cut points [31] to determine estimates of time per day spent in different intensities of activity, and there is no degree of predicting METs for users in Active style Pro. Because the time spent in vigorous PA (VPA) was low (Active style Pro; sample average of 1.5 ± 4.2 min/day), MPA and VPA were combined into one variable (MVPA) [48]. The number of steps (steps per day) were calculated from each device.

### 2.3. Statistical Analysis

The variables on work, nonwork, and total days (work day and nonwork day) were analyzed separately. All participants who achieved two valid dates (including at least one work day and one nonwork day) were selected. The differences of outcomes measuring PA and SB for each device between work and nonwork days were examined using independent *t*-tests.

The differences in outputs for total step counts and mean time of each activity intensity between accelerometers were examined using paired *t*-tests in total, work, and nonwork days, respectively. Using Pearson correlations, intraclass correlation coefficients (ICC), and Bland–Altman plots, agreement between the two devices was analyzed for total, work, and nonwork days, respectively. Bland–Altman plots were created to assess the bias and limits of agreement of total step counts and the time of each activity intensity. Bland–Altman plots simply quantify the bias and range of agreement and it is expected that 95% of the differences between two measurements should be between the mean difference ± 1.96 × standard deviation [34]. A positive mean difference suggests an overestimate and a negative mean difference suggests an underestimate of Active style Pro estimated values compared to each activity estimated by the ActiGraph using vertical axis and vector magnitude. Proportional difference was measured by Pearson correlation coefficients, which detected the proportional trend of the magnitude of the measurement. 

We used correction factors to compare the result from the Active style Pro and ActiGraph devices. Using correction factors, participants’ data were randomly assigned to validation (to provide the correction factors) and cross-validation (to examine the differences between before and after using correction factors) groups. Correction factors were calculated by drawing a random sample of 22 of the 44 participants (work day: 23 of 47; nonwork day: 22 of 45) and using linear regression models to identify significant differences between the Active style Pro and ActiGraph in time spent in SB, LPA, and MVPA. The linear regression model, which provided the regression equation with one dependent (outcomes of Active style Pro) and one independent (outcomes of ActiGraphs) variable, was defined by the formula y = c + bx, where y is the estimated dependent variable score (corresponding outcomes of ActiGraphs), c is a constant, b is the regression coefficient, and x is the score of the independent variables (outcomes of Active style Pro). 

In cross-validation groups, these correction factors were used to recalculate the corresponding results from the ActiGraph vertical axis and vector magnitude in the remaining sample (*n* = 22), which were then compared to the Active style Pro data of that subsample. The difference in results from Active style Pro and the corresponding results from the ActiGraph vertical axis and vector magnitude were estimated using paired *t*-tests to compare Active style Pro to each axial setting in ActiGraph independently. A correction factor may be helpful to compare the different studies using ActiGraph, which is most frequently used in field-based research, and Actve style Pro to see if there are differences between them.

## 3. Results

### 3.1. Participants’ Characteristics 

Those participants who had invalid or missing accelerometer data (total, work, and nonwork days, *n* = 6, *n* = 3, and *n* = 5, respectively) due to insufficient wearing time (work and nonwork days, *n* = 2 and *n* = 4, respectively) and positioning errors (work and nonwork days, *n* = 1 and *n* = 1, respectively) were excluded. 

Final data from 44 (men, *n* = 25), 47 (men, *n* = 26), and 45 (men, *n* = 26) participants were included in this study. The characteristics of the participants are expressed as mean ± standard deviation in terms of total, work, and nonwork days: age, 42.4 ± 12, 41.6 ± 12.1, and 42.2 ± 12.0 years; height, 166.4 ± 8.4, 166.0 ± 8.4, and 164.5 ± 8.3 cm; weight, 62.3 ± 11.6, 61.6 ± 11.5, and 61.3 ± 11.4 kg; and body mass index, 21.8 ± 4.6, 22.1 ± 3.1, and 22.0 ± 3.1 kg/m². Significantly different outcomes were found between work and nonwork days in wear time, LPA time, total steps for Active style Pro and ActiGraph vertical axis and vector magnitude, and MVPA time for Active style Pro (t = 2.0–4.4, df = 90, *p* < 0.05). 

### 3.2. Active Style Pro Outcomes and ActiGraph 

Table 1 shows the measuring outcomes for the Active style Pro and ActiGraph vertical axis and vector magnitude. Active style Pro measured mean values of SB, LPA, and MVPA time of 7.3–8.0 h/day, 4.2–6.0 h/day, and 1.0–1.5 h/day for total, work, and nonwork day participants, respectively. The ActiGraph vertical axis measured mean values of SB, LPA, and MVPA time of 9.0–9.2 h/day, 3.5–5.0 h/day, and 0.6 h/day for total, work, and nonwork day participants, respectively. The ActiGraph vector magnitude measured mean values of SB, LPA, and MVPA time of 7.3–8.0 h/day, 5.6–6.7 h/day, and 0.7 h/day for total, work, and nonwork day participants, respectively.

### 3.3. Comparison between Active Style Pro and ActiGraph 

Table 2 shows the mean differences of PA output between Active style Pro and ActiGraph. All intensities of PA and SB time and total steps were significantly different between the Active style Pro and ActiGraph vertical axis and vector magnitude for total, work, and nonwork day participants. Meanwhile, only SB time was not significantly different between the Active style Pro and ActiGraph vector magnitude for day classifications (total, work, and nonwork days). The Active style Pro had lower SB time compared with the ActiGraph vertical axis, with a significant difference in total, work, and nonwork days (mean difference for total, work, and nonwork days were −87.0 min, *p* < 0.01; −102.1 min, *p* < 0.01; and −70.3 min, *p* < 0.01, respectively). The ActiGraph vector magnitude measured higher LPA time than the Active style Pro and ActiGraph vertical axis. For outcomes measuring MVPA, the Active style Pro tended to provide lower estimates than the ActiGraph vertical axis and vector magnitude. In addition, the difference between the ActiGraph vertical axis and vector magnitude was much smaller than the difference between the Active style Pro and the ActiGraph vertical axis and vector magnitude. The difference in step count between the Active style Pro and the ActiGraph was significant on work days but nonsignificant on nonwork days (*p* = 0.21). 

The values from the Active style Pro and the ActiGraph using vertical axis and vector magnitude were highly correlated for all intensities of activity (r = 0.73–0.97, all *p* < 0.01; tables not shown). The ICC showed high agreement between the Active style Pro and the ActiGraph vertical axis (ICC ≥ 0.8, all *p* < 0.001) for all intensities of activity except MVPA (total: ICC = 0.68; work day: ICC = 0.57). The Bland–Altman analysis demonstrated that agreement between the Active style Pro and the ActiGraph with both settings (vertical axis and vector magnitude) revealed that there was proportional bias for SB, LPA, and MVPA time and steps on total and nonwork days and MVPA time on work days (*r* = 0.30–0.63, *p* < 0.05). These results showed that the measurement difference between the Active style Pro and the ActiGraph tended to be higher when the mean of measurement between Active style Pro and ActiGraph was larger (Figure 1 and Appendix A).

Table 3 shows the differences of the output valuables from the remaining sample between before and after applying the correction factors (Table 4). The correction factors for the different accelerometer output variables are presented in Table 4. The data were more suitable to data in the Active style Pro after the correction factors were applied: the mean difference was smaller, and the paired *t*-test showed that, for all accelerometer output variables except for LPA, the mean difference between the Active style Pro and the ActiGraph (vertical axis and vector magnitude) was not significantly different.

## 4. Discussion

The present study showed that there was no significant difference in calculated SB between the Active style Pro and ActiGraph vector magnitude in total, work, and nonwork days, while all PA intensities and steps were significantly different between the Active style Pro and the ActiGraph vertical axis and vector magnitude. The difference of SB output differed depending on the axial settings. These findings provide evidence that the ActiGraph vector magnitude has good agreement in estimating SB time with Active style Pro, regardless of the type of activity pattern (work and nonwork days), whereas the ActiGraph vertical axis measured more SB time than the Active style Pro. This result is consistent with previous studies comparing accelerometers using vertical axis with vector magnitude to assess SB time. Some studies using vertical axis settings have shown that accelerometers using vertical axis may overestimate the amount of SB [37,49,50]. For example, a study conducted in the Netherlands found an overestimation of sedentary time by almost 1.7 and 2.5 h compared with ActiGraph vector magnitude and activPAL as a criterion method, respectively, under free-living conditions [49]. Another recent study in Japan demonstrated that the ActiGraph vertical axis significantly overestimated SB compared with the Active style Pro and activPAL [37]. Thus, Active style Pro and ActiGraph vector magnitude were similarly accurate in measuring SB time for adults under free-living conditions at the group level. A study validating the identification of SB activities by two triaxial accelerometers under laboratory condition found better accuracy for sedentary activities for the ActiGraph vector magnitude and Sensewear accelerometer (BodyMedia, Pittsburgh, PA, USA) than the ActiGraph vertical axis [51]. A recent study examined the difference between the triaxial accelerometers RT3 (Stayhealthy, Inc, Monrovia, CA, USA) and Sensewear, which showed that there were no significant differences for SB times between the two devices under free-living conditions [52]. In the present study, the ActiGraph vector magnitude showed similar results in SB time (within a 2 min difference), whereas using the vertical axis in ActiGraph resulted in measuring more SB time by almost 1 h than the two accelerometers using vector magnitude. This is consistent with previous studies [51,52], even though the Sensewear accelerometer is arm-worn and concordance in the previous study is limited at the group level [52]. However, to the best of our knowledge, there are few studies comparing measured SB time for the ActiGraph vector magnitude and triaxial accelerometers in different hip-worn devises under free-living conditions. Therefore, it is helpful for public health researchers and sports scientists to accumulate evidence on SB research [53]. However, the sample in this study was limited to healthy adults. Future works may need to examine the effect in different population groups.

The most remarkable difference in measuring PA between the Active style Pro and the ActiGraph was MVPA time. The ActiGraph vertical axis and vector magnitude estimated significantly less MVPA time compared with the Active style Pro. Many studies have examined the difference between ActiGraph and other triaxial accelerometers under free-living conditions. One study in Canada showed that there was a significant difference of mean MVPA time between the ActiGraph and activPAL3 (*p* > 0.05, all) in adults (51–75 years); in terms of mean MVPA time, the ActiGraph vertical axis and vector magnitude and ActivPAL3 measured 0.68 h/day, 1.05 h/day, and 1.21 h/day, respectively [24]. Another study for participants aged 13–16 years clarified the difference between the ActiGraph vertical axis and RT3 using the vector magnitude setting in moderate and vigorous intensity PA under free-living conditions. RT3 measured greater means of moderate and vigorous time (%, 0.69 time/day and 0.02 time/day, all *p* < 0.05) [54]. Our results in this study are consistent with previous studies [24,54]. Thus, ActiGraph devices may have a tendency to measure less MVPA time than other accelerometers using vector magnitude under free-living conditions, regardless of the axial setting. Especially in comparison with the Active style Pro, it measured over 20 min/day more than both ActiGraph axial settings. The axial setting is one of the most important factors affecting PA assessment. Previous studies have suggested that accelerometers using vector magnitude provide a more accurate estimate of PA time than accelerometers using vector magnitude [55,56] because they measure more informative accelerations (i.e., anteroposterior and mediolateral accelerations) in addition to vertical accelerations. However, for the current outcome regarding MVPA, there were few effects resulting from different axial settings. Therefore, considering the PA recommendations of at least 150 min/week of MVPA [57], this difference of MVPA between the Active style Pro and ActiGraph devices should be taken into account when using different types of devices.

While the differences between the Active style Pro and the ActiGraph limit the comparability of SB and PA outcomes across studies, correction factors developed based on linear regressions can mitigate such differences [58]. The present study showed that the data were more suitable after the correction factor approach was applied: the mean difference was smaller for all accelerometer outcome variables. The high agreement between the Active style Pro and the modified ActiGraph demonstrated that the data extracted from these devices may be suitable to compare with results measuring MVPA and SB. Thus, this study improved the understanding of the differences and agreements between these two accelerometers measuring PA and SB. Since ActiGraph is most commonly used in research as a research-grade accelerometer [28,46,59], these findings will aid in comparing and interpreting evidence from studies using Active style Pro accelerometers. 

However, the correction factors did not improve the comparison for the time spent in LPA. It shows that there is an inherent difference of assessment of LPA on work (vertical axis) and nonwork (vector magnitude) days between the Active style Pro and the ActiGraph, which is affected by different patterns of activity. It has been reported that people spend much more time in LPA than MVPA [46,60,61], and recent studies have demonstrated that objectively measured LPA is associated with physical function, mental health, and mortality [62,63,64]. Thus, it may help understand these differences when investigating various types of activities under laboratory conditions to confirm whether there are different estimations of the same activities in future studies [65,66]. These findings should help researchers and sports scientist to choose an accelerometer in research measuring PA and SB.

This study has some limitations. First, since only healthy adults were recruited, the results cannot be generalized for other groups such as young people or older adults. Moreover, our participants had a small amount of time engaged in VPA. Therefore, the outcomes in high intensity activity (i.e., more than 6.0 METs) were not separately examined in this study. Second, the SB and PA estimations were not compared with a gold standard (e.g., doubly labeled water, indirect colorimetry, direct observation, or activPAL) because the aim of this study was to investigate differences between the Active style Pro and the ActiGraph during free-living activities. Additionally, ActiGraph accelerometer cannot be used as the criterion measure to assess the validity of Active style Pro. Finally, the thresholds developed by Freedson et al. [44] and Sasaki et al. [45] were chosen for the present study, although ActiGraph has numerous cut points which have been developed over two decades, and different cut points cause differences in PA and SB measurements [58]. Our findings may not be applicable to other cut points. In addition, relatively new advanced methods such as machine learning approaches and artificial neural networks have been developed to estimate the time spent in PA intensity categories for the ActiGraph device, addressing many of the problems inherent in count-based approaches. Future studies need to use these new classification methods that improve the accuracy of measuring diverse types of physical activities. 

## 5. Conclusions

The present analyses indicated that: (1) there is a significant difference between PA time assessed with the Active style Pro compared with the ActiGraph vertical axis and vector magnitude accelerometers, regardless of activity pattern; (2) the SB results from the Active style Pro and the ActiGraph vector magnitude accelerometers can be directly compared at the group level on adult populations; and (3) SB and MVPA differences between the Active style Pro and ActiGraph vertical axis and vector magnitude accelerometers were reduced using correction factors. When the ActiGraph vertical axis and vector magnitude accelerometer are used, the correction factors developed in the present study may be used to convert SB and MVPA accelerometer outcome variables to be comparable with data assessed using the Active style Pro accelerometer.

## Figures and Tables

**Figure 1 ijerph-16-03065-f001:**
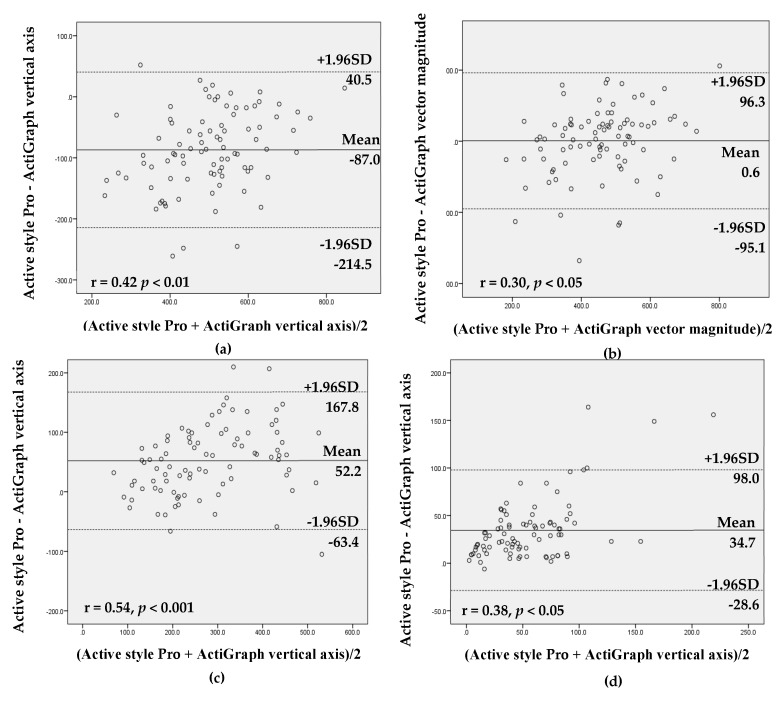
Bland–Altman analysis for the data agreement assessed with the Active style Pro and the ActiGraph vertical axis and vector magnitude (total sample including work and nonwork days): (**a**) sedentary behavior (SB) time, vertical axis; (**b**) SB time, vector magnitude; (**c**) light-intensity physical activity (LPA) time, vertical axis; (**d**) moderate-to-vigorous physical activity (MVPA) time, vertical axis; (**e**) MVPA time, vector magnitude; and (**f**) steps.

**Table 1 ijerph-16-03065-t001:** Participants’ characteristics.

Participants’ Characteristics	Total Participants	Work Day	Nonwork Day
Accelerometers’ outputs			
Wear time (min/day)	835.2 ± 108.7	911.8 ± 109.6	839.9 ± 94.3
Total time of SB (min/day)			
Active style Pro	456.4 ± 134.3	437.1 ± 132.2	480.6 ± 134.3
ActiGraph vertical axis	543.5 ± 114.5	539.2 ± 124.1	550.9 ± 108.1
ActiGraph vector magnitude	455.9 ± 120.1	435.0 ± 132.6	480.3 ± 102.4
Total time of LPA (min/day)			
Active style Pro	306.1 ± 127.4	356.9 ± 121.7	251.7 ± 108.9
ActiGraph vertical axis	253.8 ± 110.3	298.4 ± 121.4	208.6 ± 78.2
ActiGraph vector magnitude	334.9 ± 131.6	393.5 ± 134.4	274.3 ± 96.6
Total time of MVPA (min/day)			
Active style Pro	72.1 ± 49.1	81.9 ± 52.9	61.4 ± 41.3
ActiGraph vertical axis	37.4 ± 31.3	37.0 ± 28.9	36.4 ± 33.0
ActiGraph vector magnitude	43.9 ± 36.9	46.1 ± 38.5	41.3 ± 34.1
Total steps (steps/day)			
Active style Pro	8691.4 ± 4959.8	9969.0 ± 4355.1	7417.9 ± 5132.6
ActiGraph (vertical axis, vector magnitude)	8373.0 ± 4585.5	9441.8 ± 4190.9	7175.9 ± 4582.9

Note: mean values ± standard deviation, SB, sedentary behavior; LPA, light physical activity; MVPA, moderate-to-vigorous physical activity.

**Table 2 ijerph-16-03065-t002:** The mean differences of PA output between Active style Pro and ActiGraph.

	Mean Difference (95% CI)		
	Total	Work Day	Nonwork Day
Total time of SB (min/day)			
Active style Pro—ActiGraph vertical axis	−87.0 (−100.8, −73.2) *	−102.1 (−121.2, −82.9) *	−70.3 (−89.6, −50.9) *
Active style Pro—ActiGraph vector magnitude	0.6 (−9.8, 10.9)	2.1 (−10.9, 15.2)	0.3 (−15.2, 15.9)
Total time of LPA (min/day)			
Active style Pro—ActiGraph vertical axis	52.2 (39.7, 64.7) *	58.5 (40.2, 76.8) *	43.0 (25.2, 60.9) *
Active style Pro—ActiGraph vector magnitude	−28.8 (−39.3, −18.3) *	−36.6 (−51.1, −22.2) *	−22.6 (−37.0, −8.3) *
Total time of MVPA (min/day)			
Active style Pro—ActiGraph vertical axis	34.7 (27.9, 41.6) *	44.9 (34.0, 55.8) *	25.1 (18.2, 32.0) *
Active style Pro—ActiGraph vector magnitude	28.2 (22.6, 33.8) *	35.8 (27.3, 44.2) *	20.1 (14.1, 26.2) *
Total steps (step/day)			
Active style Pro—ActiGraph (vertical axis, vector magnitude)	318.3 (59.8, 576.9) *	527.2 (157.2, 897.2) **	242.0 (−141.1, 625.1)

* *p* < 0.01. Note: SB, sedentary behavior; LPA, light physical activity; MVPA, moderate-to-vigorous physical activity.

**Table 3 ijerph-16-03065-t003:** The differences of PA outputs between Active style Pro and ActiGraph using the correction factors.

	Active style Pro	ActiGraph Vertical Axis	Difference (95% CI)	*p*
Total (*n* = 44, wear time = 829.5 ± 116.6)				
SB	462.0 ± 150.1	467.5 ± 116.3	−5.5 ± 64.5	0.57
LPA	296.8 ± 134.8	292.3 ± 95.4	4.4 ± 61.5	0.64
MVPA	69.6 ± 42.4	64.6 ± 31.8	5.0 ± 30.4	0.28
Work day (*n* = 23, wear time = 880.3 ± 102.0)				
SB	464.9 ± 154.6	443.5 ± 116.5	21.4 ± 57.0	0.09
LPA	325.8 ± 125.7	355.0 ± 93.1	−29.2 ± 54.3	<0.05
MVPA	93.0 ± 63.1	90.0 ± 33.8	2.9 ± 41.5	0.74
Nonwork day (*n* = 22, wear time = 800.0 ± 95.1)				
SB	469.1 ± 103.3	475.7 ± 106.3	−6.6 ± 42.0	0.47
LPA	260.1 ± 90.1	246.3 ± 86.2	13.8 ± 40.6	0.12
MVPA	68.5 ± 42.0	67.7 ± 46.2	0.8 ± 23.6	0.87
	**Active style Pro**	**ActiGraph Vector Magnitude**	**Difference (95% CI)**	***p***
Total (*n* = 44, wear time = 829.5 ± 116.6)				
LPA	296.8 ± 134.8	300.4 ± 124.2	−3.7 (−20.1, 12.8)	0.66
MVPA	69.6 ± 42.4	65.3 ± 30.9	4.2 (−4.3, 12.8)	0.33
Work day (*n* = 23, wear time = 880.3 ± 102.0)				
LPA	325.8 ± 125.7	327.7 ± 129.0	−1.9 (−18.5, 14.6)	0.81
MVPA	93.0 ± 63.1	99.5 ± 60.3	−6.5 (−17.8, 4.7)	0.24
Nonwork day (*n* = 22, wear time = 800.0 ± 95.1)				
LPA	260.1 ± 90.1	276.5 ± 107.1	−16.4 (−32.0, −0.7)	<0.05
MVPA	68.5 ± 42.0	66.3 ± 44.7	2.2 (−6.4, 10.8)	0.60

Note: SB, sedentary behavior; LPA, light physical activity; MVPA, moderate-to-vigorous physical activity.

**Table 4 ijerph-16-03065-t004:** The correction factors based on the linear regression.

	Total (*n* = 44)		Work Day (*n* = 24)		Nonwork Day (*n* = 23)	
Outcome variables	ActiGraph vertical axis	ActiGraph vector magnitude	ActiGraph vertical axis	ActiGraph vector magnitude	ActiGraph vertical axis	ActiGraph vector magnitude
SB time (%, min/day)	−0.13 + 1.04 × SB	non sig	−0.03 + 0.84 × SB	non sig	−0.34 + 1.37 × SB	non sig
LPA time (%, min/day)	0.1 + 0.87 × LPA	0.01 + 0.90 × LPA	0.2 + 0.66 × LPA	0.01 + 0.88 × LPA	0.004 + 1.15 × LPA	0.08 + 1.18 × LPA
MVPA time (%, min/day)	0.03 + 1.14 × MVPA	0.03 + 1.10 × MVPA	0.05 + 0.94 × MVPA	0.03 + 1.20 × MVPA time	0.02 + 1.22 × MVPA	0.02 + 1.17 × MVPA

Note: SB, sedentary behavior; LPA, light physical activity; MVPA, moderate-to-vigorous physical activity. Corresponding ActiGraph time = AG each intensity time (min, min/day); Intercepts × Wear time (min, min/day) + Slopes × ASP each intensity time (min, min/day).

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
