# Peer review of "Assessing Physical Activity and Sedentary Behavior under Free-Living Conditions: Comparison of Active Style Pro HJA-350IT and ActiGraphTM GT3X+"

_ijerph, 2019, doi:10.3390/ijerph16173065_

Round 1
Reviewer 1 Report
Reviewer’s comment:
The authors did a good job in responding to most of two other Reviewers comments. I am satisfied with the way you have improved the manuscript and answered the reviewers comments. However, there are a few concerns that remain to be addressed.
I still think ActiGraph accelerometer cannot be used ‘gold standard’ as the criterion measure to assess the validity of Active style Pro. Moreover, activPAL is better than ActiGraph to estimate sedentary behavior. This information Page 12 Line 35-47 would be have been very useful to state this problem in the methods section. However, I suggest that the author would to discussion more in the limitation part in this study.
Do you have energy expenditure (kcals) data the mean differences of PA output between Active style Pro and ActiGraph? What is ICC for vigorous PA between the two devices? The authors have not addressed this. Please provide the reader with information about vigorous physical activity (VPA) information. I suggest the writing of table 2-4 separated the SB and PA results, in PA results add energy expenditure (kcals) and VPA data.
Reviewer 2 Report
Nothing
Author Response
Please see the attachement.

This manuscript is a resubmission of an earlier submission. The following is a list of the peer review reports and author responses from that submission.
Round 1
Reviewer 1 Report
General remarks
The study compares the estimates for physical activity (low, moderate and vigorous) and sedentary behaviour of two research sensors: The ActiGraph GT3X+ (THE established sensor in this field) and the Active style Pro HJA-350IT (a less known and less common sensor). This is an interesting topic, aiming to make the results of two sensors comparable, and thus allowing to compare different studies conducted all over the world. However, the manuscript has some substantial weaknesses that require a very detailed revision. I recommend the authors to careful check whether they are able to address the subsequent criticisms before they start to revise the manuscript. In a general sense, the manuscript
1) lacks a clear focus
2) needs a proper language editing (in particular singular and plural, time use, word order) including a more precise choice of words (remove layman expressions like “more comparable”, “more informative accelerations”)
3) lacks essential information (e.g. recording frequency, non-wear time detection) while other information are repeated several times or confusingly explained (e.g. wear time)
4) The presented results and discussion are not in line with the study aim
5) The discussion is hard to follow, includes a lot of guessing and theoretical arguments not investigated in this study. Instead, a true discussion of the study findings would strengthen the manuscript content.
Abstract:
Please reorganize the abstract to give it a clear structure, in particular for the result section (results appear to be listed randomly). Please take only the key findings to help the reader to figure out what is most important.
Add a method section (e.g. how the correction factors were developed, describe data processing and analysis).
The abstract and the entire text body needs a proper English editing (e.g. line 19: have instead of has, line 20: work-days and non-work days (plural))
line 22: what is meant with “each day of a work-day and a non-work day”?
Consider to replace uni-axis with vertical axis, and tri-axis with vector magnitude in the entire manuscript (if this is meant).
Introduction
Page 2, Line 5: Please rephrase: “The advantage of this approach [objective measurement] include objective measurement of PA….” makes no sense.
Line 12: consider to remove: in the literature. It’s not the literature that uses these sensors.
Line 14: What is meant by “sensitivity” and “wearing-time” as a feature of each device?
Line 16 gives an example for the comparison of ActiGraph and Actical, but the example is introduced with respect to device generation. Please clarify.
Line 32: Mention that the high validity is only valid for the lab (as I assume since it is compared to the Douglas bag method).
Line 33-34: This statement (“Precise assessments of LPA and SB are the key advantages of Active style Pro”) needs a very strong evidence, more details (under which circumstances), and must be referenced if it is actually true. Why not for MVPA?
Line 37-39: If the aim of this study is to compare LPA and MVPA, please remove SB from the evaluation. Or give also the reasoning why SB is an aim of this study. The entire result and discussion section should be more sharpened towards the study aim.
Line 38: confusing with the two different tri-axial. Could vertical axis and vector magnitude solve the confusion?
Material and Methods
Line 50: a total of two days were recorded, but later a wear time of ≥4 days (Page 3, line 4-5) is used as inclusion criteria. Please explain this gap.
Page 3, Line 1-2: Already mentioned previously that these two sensors were worn (Page 2, Line 48-49). In general for the entire manuscript: Please mention the facts only once, unless they are really needed to increase the readability of the text.
Line 11: the company name for each sensor should be mentioned only once.
Line 14: Did the Active style Pro estimated the intensity every 60 seconds (one point in time), or did it average the intensity for 60 second époques?
Line 16: Unclear whether the locomotion vs. non-locomotion algorithm was used in the present study.
Line 19-20: Please use the same units here as for the Active style Pro.
Line 23-25: instead of giving those fundamental sensor properties, it would be of help to read here about the sensor properties used in this study.
Line 27-28: How did you clarify which setting is more suitable to compare? The one that performed better?
Line 30-33: Please give some details of data processing (e.g. low-frequency extension filter, non-wear time detection).
Line 30-33 repeats 25-27. Consider to merge these parts to increase the readability.
Line 34: Surprised to read here about sedentary behaviour (not in line with the study aim).
Line 34 to 45: It might be easier to follow if it is first explained what is going to be measured, and second how this is done with each of the sensor.
Line 36: in case sedentary behaviour is defined solely by energy expenditure, consider the use of minimal Physical Activity as outlined by Holtermann and colleagues (doi: 10.1016/j.apergo.2017.03.012)
Line 36-38 repeats again line 30-33 and line 25-27. Keep this information together, and remove double mentions. It’s ugly to read.
Line 38-39 repeats 34-35, and line 43-44 adds the next piece of information. Please remove those jumps from topic to topic.
Line 39: the table indicates for the Active style Pro the same information than in the text (doubling). Is there no other information available for the Active style Pro (e.g. cut-points?). Table 1 presents 6 new numbers, not sure whether this justifies the use of a Table. The information could be straightforward included in the text.
Line 41-42: not sure whether it is a good idea to merge two categories just because one category has small numbers.
Line 48: What is meant with Total participants had valid dates in both of work day and non‐work day?
Line 50 – page 4, line 1: Which statistical measure was used for what? Please indicate why you are using ICC, correlation, ANOVA, and Bland-Altman? And why do you not use the same statistical measures for the correction factors (here, scatter plots with identity lines are used)? On which level were the numbers compared (day by day? subject by subject?)
Page 4, line 4: how were these correction factors developed? Why were they calculated for the ActiGraph, the sensor most frequently used, and not for the far less common Active style Pro? Is there a specific reasoning?
Line 11: The line has a T-test?
Results
Line 17: What is the reason for invalid and missing data?
Line 20, Table 2: Is there a specific need to present all those numbers for total, work and non-work days separately? The sample is not that different. Most information in table 2 is not explained or discussed, it just stands there. You might first check whether there is a difference between work and non-work days, and present only the combined numbers if there is none.
Table 2: Is total steps actually presented as minute per day?
Page 5, Line 1-18: I can not follow your thoughts through this section. The results seem to appear randomly, and lacks any kind of structure. Maybe use headings for the sensor comparison, and sub headings for SB, LPA, and MVPA. Or headings for each behaviour, and then explain the findings in a structured way.
Line 2 says that the values were highly correlated, while line 16 says that the agreement was not acceptable. If the method section describes why each measure was used, it might get easier to follow these results.
The method section says that ANOVA and t-test were also used. Are these results presented in this section?
Line 8-10 repeats line 5, and in between, a different comparison is made. Hard to follow.
Line 11 to 12 repeats line 4-5. Again, it is hard to figure that out.
Line 14: With so many comparisons made, did you take multiple comparison into account? (e.g. Bonferroni)
Line 17: is it relevant that there is no “proportional bias” if the agreement is not acceptable? How was the “proportional bias” assessed?
Page 6, Table 3: The two different ActiGraph methods are also compared. Please include this comparison in the study aim, or remove it from the manuscript.
Page 7, Figure 1: It is not a good idea to distribute figures (and tables, e.g. number 5) over several pages.
Figure 1: There are more dots in the figure than subjects (It seems that the result for work and non-work day of each subject are included twice). This artificially increases the sample size. Please use only one “dot” per subject
Figure 1 only includes the vertical axis, why not the vector magnitude?
Page 9, line 1: Table 4 shows no correction factors, maybe you mean table 5?
Line 2: What is meant with more comparable?
Page 10, Table 5: why is sometimes wear time included and sometimes not? Remove all those repetitions to make the table readable (e.g. each cell in column 2 says “ActiGraph vertical axis”, is that necessary?). It is unclear how those numbers were generated, and how they shall be used.
Page 12-14, Figure 2: what is collection? Which units are used? In plot a, it seems that there is a “proportional bias” in the data (ActiGraph higher for low numbers, and lower for high numbers). Is this mentioned somewhere? The collection (should this be correction?) does not take this into account, for a particular reason?
Discussion
Page 15, line 2: Please explain why there should be no difference although the agreement was “not acceptable” (page 5, line 15-16).
Line 5-6: Why can the sensors be directly compared if the output differs (line 4)?
Line 8-11: Does the cut-off discussion have anything to do with the present study?
Line 11-13: Why not discussing your own findings in relation to this other study. Are the results the same?
Line 16-17: is this guessing, or can you substantiate this statement with evidence?
line 28-30: there are multiple regression approaches available for the ActiGraph.
Line 23-37: This section seems to argue that the Active style Pro provides more accurate estimates for MVPA than the ActiGraph, but the current study does not provide any relevant information to this topic. As mentioned in line 21, a valid reference criteria should be used to answer this question. Accordingly, I would remove the entire section and instead discuss the results of the present study.
Line 41-42 says that the axial setting “may cause” the difference. Since this is the only difference between the vector magnitude and vertical axis processing, what else “could” cause this difference? The difference should be for sure caused by the axial setting.
Line44: What are more informative accelerations?
Page 16, line 4: If the sensors are interchangeable, I would expect to have very narrow limits of agreement for each single subject. However, with the presented results, I can only speculate about it, and it doesn’t look like.
Line 6-7: I don’t think that both are widely used in research, are they?
Line 9-10: what means “The difference between Active style Pro and ActiGraph cut points is based on the main vertical axis and vector magnitude.”?
Line 15-16: Please explain why there is a need for laboratory studies.
Line 24: Why do we need new studies using other thresholds? Other thresholds could be easily analysed with your dataset.
Line 28-29: directly compared on group level (but not on an individual level).
Line 29-30: How can you remove the difference between SB estimates from Active style Pro and ActiGraph vector magnitude with correction factors if there is none according to point 2?
Reviewer 2 Report
My comments/questions are as follows:
Generally, we use a more precise device or use gold standard as the criterion measure to assess the validity of an accelerometer or pedometer.
To assess the validity of the new accelerometer, we should choose indirect calorimetry to evaluate validity of Active style Pro (ASP) or ActiGraph (AG) accelerometer. If we estimate validity of a new questionnaire, we can use an accelerometer to figure out the ICC agreement between the accelerometer and the questionnaire.
Therefore, it is uncommon to compare two accelerometer to figure out validity of accelerometer.
Reviewer 3 Report
Improve the results and tables presented
